# In Vitro Biological Activity and In Vivo Human Study of Porcine-Placenta-Extract-Loaded Nanovesicle Formulations for Skin and Hair Rejuvenation

**DOI:** 10.3390/pharmaceutics14091846

**Published:** 2022-09-01

**Authors:** Kritsanaporn Tansathien, Tanasait Ngawhirunpat, Worranan Rangsimawong, Prasopchai Patrojanasophon, Praneet Opanasopit, Nopparat Nuntharatanapong

**Affiliations:** 1Pharmaceutical Development of Green Innovation Group (PDGIG), Faculty of Pharmacy, Silpakorn University, Nakhon Pathom 73000, Thailand; 2Division of Pharmaceutical Chemistry and Technology, Faculty of Pharmaceutical Sciences, Ubon Ratchathani University, Ubon Ratchathani 34190, Thailand

**Keywords:** niosomes, nanovesicles, porcine placenta extract, skin repair, hair growth promotion

## Abstract

Porcine placenta extract (PPE) contains many water-soluble macromolecular compounds, such as proteins and growth factors, which have limited transportation through the skin. This study aimed to assess the effect of porcine-placenta-extract (PPE)-loaded nano-transdermal systems for skin repair and hair growth promotion. The potentials of the nanoformulation for cytotoxicity, cell proliferation, intracellular reactive oxygen species (ROS) reduction, lipoxygenase inhibition, intracellular inflammatory cytokine reduction, and cell aggregation were evaluated. PPE-entrapped niosome nanovesicles were produced by thin-film hydration and probe-sonication methods, followed by incorporation in a skin serum formulation. The physicochemical properties of the formulation were examined, and the efficacy of the serum formulation was elucidated in humans. The results showed that PPE had no toxicity and was able to induce cell growth and cell aggregation. In addition, PPE significantly decreased intracellular ROS, inhibited lipoxygenase activity, and reduced the production of intracellular tumor necrosis factor-α. In the in vivo human study, the PPE nanovesicles-loaded serum could improve skin properties by increasing skin hydration. Moreover, it was capable of promoting hair growth by increasing hair elongation and melanin index after application for one month. Consequently, the PPE nanovesicles-loaded serum was effective for skin anti-aging and hair rejuvenation.

## 1. Introduction

The aging process of skin and hair correlates to oxidative stress and chronic inflammation, which may cause the loss of skin and hair functions [1]. Exposure of the skin to external physical factors, for example, chemicals and ultraviolet (UV) rays, can cause the skin to produce reactive oxygen species (ROS) [2]. Imbalanced redox homeostasis may result in dysregulation of redox-sensitive signal transduction pathways and high levels of lipid peroxidation products, causing cytotoxic effects [2,3]. Fatty acids are essential in maintaining the functions of the epidermis. Moreover, polyunsaturated fatty acids are generated from the complex membrane or stored lipids and converted into lipid intermediates to produce inflammatory effects [4,5].

Hair aging affects the physical looks and self-perception of humans. For the hair growth cycle, the growth of hair follicles (HF) can be categorized into various stages, including stages of rapid growth and hair shaft creation (anagen), apoptosis-induced regression (catagen), and relative HF quiescence (telogen). Age-related progression of hair loss may be presented by a shortened anagen phase and persistent telogen phase [6]. Additionally, hair aging is correlated to the weathering of the hair shaft and progressive degeneration of hair fiber, including decreasing hair production. Moreover, extrinsic factors (smoking, UV radiation, etc.) can induce alopecia [7,8].

The placenta is a natural reservoir of numerous biologically active substances, including vitamins, amino acids, peptides, growth factors, trace elements, etc. [9]. For pharmacological activities in skin aging, peptides and growth factors (GFs) affect collagen breakdown and production in the skin [10]. Moreover, a previous report showed that placental extract at a sufficient concentration could preserve the anagen phase and the growth of hair cells [11].

In transdermal delivery, most hydrophilic macromolecules cannot pass through the skin by simple diffusion or passive transportation. Recently, many techniques have been investigated to overcome the skin barrier and enhance skin permeation, for example, formulation optimization, the use of physical and chemical enhancers, and stratum corneum ablation [12]. The utilization of nano-based systems is one of the most-studied approaches for macromolecule delivery, in which niosome nanovesicles are found to be beneficial for delivering hydrophilic compounds due to nontoxicity, low production costs, and higher chemical stability [13]. Sponge spicules are micron-sized needles that can physically disturb the stratum corneum, leading to the improved skin delivery of hydrophilic active macromolecules [14]. Recently, we successfully developed an innovative formulation containing sponge spicules for increasing the skin permeation of macromolecules using a deer antler velvets (DAVs) extract [15]. However, the activities of porcine placenta extract for reducing skin and hair anti-aging and the effect of a PPE-entrapped niosome-nanovesicles-serum-containing sponge spicules for improving skin conditions in humans have not been studied. Therefore, this study aimed to examine the bioactivities of porcine placenta extract in reducing skin and hair aging and to evaluate the efficacy of PPE-entrapped niosome-nanovesicles-serum-containing sponge spicules in improving skin conditions in human subjects. 

## 2. Materials and Methods

### 2.1. Materials

Dulbecco’s eagle medium (DMEM), 0.5% trypsin-EDTA, Glutamax^TM^, non-essential amino acid, and penicillin–streptomycin were procured from Gibco, Grand Island, NY, USA. Fetal bovine serum (FBS) was obtained from Gibco, Paisley, PA4 9RF, UK. 2’,7’-dichlorodihydrofluorescein diacetate (H_2_DCFDA) was provided by Invitrogen, Thermo Fisher Scientific, Eugene, OR, USA. 3-(4,5-Dimethyl-2-thiazolyl)-2,5-diphenyl-2H-tetrazolium bromide (MTT), tert-Butyl hydroperoxide solution (t-BHP), lipoxidase from Glycine max (soybean), linoleic acid, intracellular tumor necrosis factor (TNF)-α enzyme-linked immunosorbent assay (ELISA) assay kit, ProteoSilver™ Silver Stain Kit, Span 20, and cholesterol were bought from Sigma-Aldrich, St. Louis, MO, USA. Sponge spicules powder (MS; 98% plus spicule; 100–180 mesh) was purchased from Hunan Sunshine Bio-Tech Co., Ltd., Changsha, Hunan, China. Normal human foreskin fibroblast cells (NHF), human keratinocytes cells (HaCaT), and human follicle dermal papilla cells (HFDPC) were attained from the American-Type Culture Collection (ATCC) (Rockville, MD, USA). The serum base was a gift from Zen Innovation Group Co., Ltd., Pathum Thani, Thailand.

### 2.2. Preparation of Porcine Placenta Extract (PPE)

The fresh porcine placenta was obtained from Charnchai Farm, Ratchaburi, Thailand. It was collected from large-white x landrace hybrid sows 2 years old with a natural delivery. The pig was vaccinated following the recommended vaccination program and was grown in standard sanitary conditions. Commercial hybrid lines of large white pigs and Landrace are common in Thailand as they have high-quality lean meat and provide a high total number of piglets born, piglets born alive, average birth weight, and average weaning weight [16]. The fresh porcine placenta was cleaned with 0.9% sodium chloride and cut into small pieces. Afterward, the placenta was blended with phosphate-buffered saline (PBS) pH 7.4 in a blender at a placenta to PBS ratio of 1:1. The grided placenta was then blended with purified water at a ratio of 2:13, and the bioactive compounds were extracted using a probe sonicator (Vibra-Cell^TM^ Ultrasonic Processor Model VC505, SONICS & MATERIALS, Inc., Newtown, CT, USA). Afterward, the mixture was centrifuged at 4000 rpm, 4 °C for 15 min. Then, the supernatant was gathered and lyophilized (FreeZone 2.5 Liter Benchtop freeze-dry system, Labconco, Kansas, MO, USA).

### 2.3. Quantification of Proteins and Growth Factors in the Extract

The total protein content in the extract was measured by bicinchoninic acid (BCA) protein assay kit (Pierce, Burlington, MA, USA) and with reference to bovine serum albumin (BSA). Other proteins were determined using 12% sodium dodecyl sulfate-polyacrylamide gel electrophoresis with an electric field of 100 mV for 90 min and stained by silver staining (ProteoSilver™ Silver Stain Kit). The contents of insulin-like growth factor-1 (IGF-1), epidermal growth factor (EGF), basic fibroblast growth factor (FGF), and transforming growth factor-β1 (TGF-β1) were measured by ELISA kits (Abcam, Cambridge, MA, USA).

### 2.4. Determination of Cell Viability

HaCaT cells (5 × 10^3^ cells/well) and NHF and HFDPC cells (1 × 10^4^ cells/well) were grown in 96-well microplates for 48 h (HaCaT) or 24 h (HFDPC) at 37 °C, 95% humidified air/5% CO_2_. The cell viability was examined by an MTT assay. Briefly, the cells were exposed to various concentrations (0–4000 µg/mL) of PPE for 24 h. Then, the medium was discarded, and the cells were rinsed with PBS at a pH of 7.4 before the addition of 125 µL of MTT reagent (0.5 mg/mL) to the wells and further incubation for 3 h. After that, the MTT solution was replaced with 100 µL of dimethyl sulfoxide (DMSO) solution to dissolve the formazan crystals in the cells. The results were read with a microplate reader (VICTOR Nivo^TM^, PerkinElmer, UK) at 550 nm, followed by calculating the percentage of cell viability according to Equation (1).
(1)$$\%\;Cell\;viability\;or\;proliferation=\frac{Absorbance\;of\;samples}{Absorbance\;of\;control}\;\;\times \;100$$

### 2.5. Determination of Cell Proliferation

NHF, HaCaT, or HFDPC cells (5 × 10^3^ cells per well) were cultured onto 96-well microplates and incubated for 48 h (HaCaT) and 24 h (NHF or HFDPC) at 37 °C, 95% humidified air/5% CO_2_. The cells were exposed to PPE (0–2000 µg/mL) for different time intervals (24–72 h). The cell proliferation was assessed by the MTT assay, as described above. The absorbance was measured using a microplate reader set at 550 nm, and the percentage of cell proliferation was computed according to Equation (2).

### 2.6. Determination of Cell Migration

NHF cells (2 × 10^5^ cells per well) were grown in cell culture plates until 70–80% cell confluence. Scratching lines were dragged from one side to the other side by a small pipette tip. PPE at a concentration of 1000 ug/mL was placed on the cells, and the cell movement behavior was observed. Serum-free DMEM was exploited as the negative control, while the positive control was the TGF-β1 standard solution (53 pg/mL). The photograph of cells was captured using an inverted microscope (Nikon^®^ T-DH, Nikon, Tokyo, Japan) at different periods.

### 2.7. Determination of Cell Aggregation of HFDPC

The HFDPC cells (8 × 10^3^ cells) were grown in 24-well plates at 37 °C with 95% humidified air/5% CO_2_. After that, the medium was removed before the cells were exposed to 1000 µg/mL of PPE for 72 h. The results were observed, and the cell-aggregation manners were photographed under an inverted microscope (Nikon^®^ T-DH, Nikon, Tokyo, Japan).

### 2.8. Determination of Intracellular ROS Reduction

In this experiment, we used H_2_DCFDA as a fluorescent probe to determine intracellular ROS in living cells according to a modified method from Wadkhien et al. (2018) [17]. HaCaT cells (2.5 × 10^4^ cells per well) were added onto 96-well black microplates and incubated for 48 h. Afterward, the medium was replaced by various concentrations (0–2000 µg/mL) of PPE or 100 µM of quercetin as a positive control and incubated for an hour. Then, 1 mM of t-BHP was added to each well and further incubated for another hour. Then, the supernatant was replaced by 20 µM H_2_DCFDA and the cells were kept at 37 °C for 30 min. The quantity of ROS production was detected by a fluorescent microplate reader (VICTOR Nivo^TM^ Multimode Plate Reader, PerkinElmer, Germany) at 485/535 nm (excitation/emission), followed by calculating the percentage of intracellular ROS reduction according to Equation (2).
(2)$$\%\;Intracellular\;ROS=\frac{Fluorescent\;intensity\;of\;treated\;cells}{Fluorescent\;intensity\;of\;non-treated\;cells}\;\times \;100$$

### 2.9. Determination of Lipoxygenase (LOX) Inhibition

In this study, LOX inhibition was investigated using a modified method from a study by Chung et al., 2009 [18]. Briefly, 50 µL of 167.5 unit/mL LOX in Tris buffer pH 7.4 was mixed with 60 µL of the test sample (PPE or standard inhibitors) and stored at 25 °C for 5 min. Water was used in place of the test sample as a control. Background blank was composed of LOX, buffer-containing water, and substrate (linoleic acid, LA). Afterward, 50 µL of LA (final concentration, 30 µM) was added and stored at 25 °C for 20 min in the dark. Finally, 240 µL of ferrous oxidation−xylenol orange (FOX) reagent was included in each centrifuge tube at 25 °C for 30 min. The results measured the Fe^3+^-dye complex from the altered colors using a microplate reader at 590 nm. The percentage of LOX enzyme inhibition was calculated following Equation (3).
(3)$$\%\;LOX\;inhibition=\frac{Absorbance\;of\;control-Absorbance\;of\;sample}{Absorbance\;of\;control}\;\;\times \;100$$

### 2.10. Intracellular TNF-α Determination

TNF-α is an inflammatory cytokine that can be produced in HaCaT cells. Briefly, 1 × 10^5^ cells per well of HaCaT cells were seeded onto 24-well microplates and incubated for 96 h at 37 °C, 95% humidified air/5% CO_2_. Various concentrations of PPE (0–1000 µg/mL) or standard inhibitor (100 µM quercetin and 300 µM aspirin (ASA)) were added to the cells and incubated for 1 h. Afterward, the cells were incubated with 1 mM t-BHP for another 1 h. Finally, the cell pellets were collected to determine the quantity of TNF-α using an ELISA assay kit.

### 2.11. Preparation of PPE-Loaded Niosome Nanovesicles and Serum Formulations

Niosomes were formulated using the thin-film hydration technique with size reduction by probe sonication. Briefly, Span 20 (1.25 mM) and cholesterol (1.25 mM) at a ratio of 1:1 were added to a glass tube containing the solvent, followed by solvent removal by purging with nitrogen gas; then, the mixture was kept in a desiccator for over 6 h to completely dry. After that, 1000 µg/mL of PPE was added and spun by a vortex machine to form the vesicles, followed by reducing the size with a probe-sonicator (Vibra-Cell^TM^ Ultrasonic Processor Model VC505, Sonics & Materials, Inc., Newtown, CT, USA) on an ice bath for 30 min (2 cycles). Excess materials were precipitated by centrifugation at 15,000 rpm, 4 °C for 15 min, and the supernatant was gathered. The PPE-loaded niosomes were mixed with a serum base composed of ethylenediaminetetraacetic acid (EDTA 2Na), glycerin, Microcare PHC, Sepimax ZEN^TM^, far-infrared (FIR) water, and sponge spicules, as previously reported [15].

### 2.12. Characterization of PPE-Loaded Niosome Nanovesicles

After complete preparation, the physicochemical properties of the PPE-loaded niosomes were determined. The particle size, polydispersity index (PDI), and surface charge were examined in triplicate (*n* = 3) at room temperature using a Zetasizer (Zetasizer Nano- ZS, Malvern Instrument, Worcestershire, UK).

Additionally, the % entrapment efficiency (*%EE*) and % loading capacity (*%LC*) of the PPE-loaded niosomes were evaluated. Briefly, the PPE-loaded niosomes were added to a centrifugal filter (Amicron^®^ 100K, Merck Millipore, Carrigtwohill, Co., Cork, Ireland), and then the filter was spun by centrifugation at a speed of 4000 rpm and a temperature of 4 °C for 15 min. The PPE-loaded niosomes were gathered, and the protein content was measured using a BCA protein assay kit (EMD Millipore, MA, USA), following the manufacturer’s instructions. The *%EE* and *%LC* were computed following Equations (4) and (5), respectively.
(4)$$\%\;EE=\frac{niosomes\;encapulated\;PPE}{Initial\;PPE\;loaded}\;\;\times \;100$$
(5)$$\%\;LC=\frac{niosomes\;encapsulated\;PPE}{Total\;content\;of\;niosomal\;materials}\;\;\times \;100$$

### 2.13. In Vivo Human Study

An in vivo human study was approved by the Investigational Review Board (Human Studies Ethics Committee, Silpakorn University Research, Innovation and Creativity Administration Office; REC63.0413-028-1705). Healthy volunteers (25–40 years old) who corresponded to the admission criteria are included in the study. The volunteers did not apply any moisturizing product to their scalps for at least 12 h before the test. Briefly, the hair at an occipital head (1 cm^2^) of the human volunteer was cut off using a razor blade. Then, the serum was applied and lightly rubbed onto the position using forefinger for 2 min twice a day, morning and evening. Skin and hair conditions (skin hydration, melanin content, hair length, and skin erythema) were measured using a DermaLab^®^ series (SkinLab Combo, Cortex Technology, Hadsund, Denmark) and a Dino-Lite Edge digital microscope (AM7915 Series, Taiwan) after continuous application of the serum for two weeks and a month. Changes in the skin condition are expressed by the values of % skin hydration and % erythema index (*%EI*), and the values were calculated according to Equations (6) and (7), respectively.
(6)$$\%\;Skin\;hydration=\frac{Skin\;hydration\;of\;test\;area-Skin\;hydration\;of\;control}{Skin\;hydration\;of\;control}\;\;\times \;100$$
(7)$$\%\;EI=100+\frac{Erythema\;value\;of\;test\;area-Erythema\;value\;of\;control}{Erythema\;value\;of\;control}\;\;\times \;100$$

The changes of hair in % melanin index (*%MI*) and % hair elongation were calculated as Equations (8) and (9), respectively.
(8)$$\%\;MI=100+\frac{Melanin\;value\;of\;test\;area-Melanin\;value\;of\;control}{Melanin\;value\;of\;control}\;\;\times \;100$$
(9)$$\%\;Hair\;elongation=\frac{Hair\;length\;of\;test\;area-Hair\;length\;of\;control}{Hair\;length\;of\;control}\;\;\times \;100$$

### 2.14. Statistical Analysis

Data are displayed as mean ± standard deviation (SD). In vitro experiments were performed with 3 replications (*n* = 3). The differences between groups were determined by one-way ANOVA and Tukey’s post hoc test. The in vivo human study was evaluated using the Wilcoxon signed-rank test. Significance differences were evident at *p* < 0.05 and *p* < 0.001.

## 3. Results

### 3.1. PPE Extract

PPE had a brown-red fibrous texture (Figure 1A), and the amount of protein was 171.82 ± 3.40 mg/g extract. SDS-PAGE displayed bands of water-soluble proteins with the MW in the range of 6.5 kDa to 66 kDa according to the protein markers. The relative molecular weight of unknown proteins and polypeptides was calculated from the standard curve of log molecular weight and relative mobility (Rf) of the protein markers. The predominant molecular weight of the unknown (unknown 1–5) of PPE on the SDS-PAGE gel were 7.20 kDa, 23.14 kDa, 32.25 kDa, 56.09 kDa, and 64.21 kDa, respectively (Figure 1B). The contents of growth factors of EGF, FGF, IGF-1, and TGF-β1 were 5.7 ± 2.47 pg/g, 65.34 ± 6.54 ng/g, 40.38 ± 6.00 ng/g, and 55.20 ± 1.21 pg/g extract, respectively.

### 3.2. Effect of PPE on Skin and Hair Cell Viability

The cell viability of NHF, HaCaT, and HFDPC cells was analyzed by MTT colorimetric assay. As shown in Figure 2A,C, NHF and HFDPC cells tended to increase cell growth after treatment with high concentrations of PPE (1000–4000 µg/mL). On the other hand, the cell viability of HaCaT was reduced at high PPE concentrations (1000–4000 µg/mL) due to the direct toxicity of the PPE toward HaCaT cells (Figure 2B).

### 3.3. Effect of PPE on Skin and Hair Cell Proliferation

The growth of NHF cells tended to increase with time after treatment with increasing concentrations of PPE. However, treating the cells with 2000 µg/mL PPE for 72 h tended to reduce cell growth (Figure 3A). PPE significantly stimulated the growth of HaCaT cells compared to the control (*p* < 0.05), especially at high PPE concentrations (1000–2000 µg/mL). Increasing the treatment time (24–72 h) led to a further increase in cell growth compared with the control (*p* < 0.05) (Figure 3B).

In terms of the effect of PPE on the hair cells, the % cell viability of HFDPC cells increased (Figure 3C) after treatment with PPE, suggesting that the PPE was able to increase cell proliferation of HFDPC cells in a concentration-dependent manner (0–2000 µg/mL), and significant cell growth was observed at high PPE concentrations (1000–2000 µg/mL) compared to the control (*p* < 0.001, *p* < 0.05). However, a further increase in the treatment time from 24 h to 72 h did not result in a further increase in cell proliferation. Based on the cell viability and proliferation studies, PPE at a concentration of 1000 µg/mL was selected for further experiments.

### 3.4. Effect of PPE on Skin Cell Migration Activity

The effect of PPE (1000 µg/mL) on cell migration is presented in Figure 4. The scratched area was narrowed at 24 h and closed at 48 h. Interestingly, PPE exhibited a higher potential to induce the migration of NHF compared to the standard TGF-β1 (positive control) and the negative control at 24 h. This result proved that PPE had an excellent ability to induce cell migration.

### 3.5. Effect of PPE on HFDPC Aggregation

HFDPC in DMEM as a control group showed the cells growing as a monolayer (Figure 5A). In comparison, in the cells treated with 1000 µg/mL PPE, the cells became dense and aggregated (Figure 5B).

### 3.6. Effect of PPE on Intracellular ROS Reduction

As shown in Figure 6, ROS production in DMEM-treated HaCaT was arbitrarily defined as 100% (control group). The exposure of the cells to 1mM t-BHP significantly increased %ROS production to 180.58 ± 9.15%. However, after the t-BHP-treated cells were incubated with PPE or quercetin, the values of % ROS production were significantly decreased (*p* < 0.05). In addition, incubating HaCaT cells with 100 µM quercetin before treatment with t-BHP at a concentration of 1 mM exhibited a % ROS production of 89.90 ± 19.30. A significant decrease in %ROS production of the cells was also observed when the cells were pre-incubated PPE before treatment with 1 mM t-BHP compared with t-BHP treated cells without pre-treatment (100%).

### 3.7. Effect of PPE on Lipoxygenase Inhibition

The lipoxygenase enzyme (LOX) inhibiting effect of PPE extract was concentration-dependent, as presented in Figure 7. PPE had a potent inhibitory potential toward LOX, with an IC50 value of 185.9 ± 2.38 μg/mL.

### 3.8. Effect of PPE on Intracellular TNF-α Reduction

The effect of PPE on intracellular TNF-*α* levels is shown in Figure 8. The amount of TNF-α secretion in untreated HaCaT cells (negative control) was 283.24 ± 19.08 pg/mg. The cells treated with quercetin (100 µM) and ASA (300 µM) (positive control) had a TNF-α content of 267.32 ± 46.97 pg/mg and 222.28 ± 52.08 pg/mg, respectively. The amount of TNF-α secretion in HaCaT cells treated with PPE (100, 200 and 1000 µg/mL) was 292.64 ± 37.82 pg/mg, 195.07 ± 16.63 pg/mg, and 211.06 ± 26.01 pg/mg, respectively. The TNF-α contents of the cells treated with 200 and 1000 µg/mL PPE were significantly decreased compared to the negative control (*p* < 0.05).

### 3.9. Characterization of PPE-Loaded Niosomal Serum

The PPE-loaded niosomal serum formulations had a light brown color due to the presence of PPE. The niosomes were nano-sized with a narrow PDI and negative zeta potential (Table 1). The *%EE* and *%LC* values of the niosomal formulation were determined before the niosomes were incorporated into the serum formulation. The niosomal vesicles could encapsulate 1000 µg/mL PPE with the *%EE* and *%LC* of 42.68 ± 2.49% and 45.33 ± 2.65%, respectively.

### 3.10. In Vivo Human Study

The effectiveness of the PPE niosome-loaded serum formulation was evaluated in human volunteers after applying the formulation to the marked area for two weeks and a month. For hair regeneration (Figure 9A), the serum-containing PPE was able to stimulate hair growth after application in all the test periods. For % hair elongation, the volunteers had significantly longer hair growth after a 1-month application (55.73 ± 14.70%) compared to the two-week application (23.65 ± 15.38), *p* < 0.05. As can be seen in Figure 9B, *%MI* significantly increased (*p* < 0.05) after treatment with the formulation for two weeks and 1 month (114.91 ± 7.59% and 135.15 ± 13.40%) compared with day 0 (102.44 ± 4.15%). This may be attributed to the growth of hair follicles because the hair follicle comprises melanin. For skin hydration (Figure 9C), the serum could enhance the skin hydration at two weeks (32.66 ± 28.95) and 1 month (30.45 ± 34.07) after application compared to day 0. However, the data at two weeks and 1 month were not significantly different. Furthermore, *%EI* significantly decreased (*p* < 0.05) after application of the serum formulation for 2 weeks (92.19 ± 7.44%) and 1 month (84.68 ± 4.72%) compared with day 0 (103.87 ± 9.43%) (Figure 9D).

## 4. Discussion

Environmental factors play an essential role in the stimulation of the aging process. The hallmarks of aging are associated with biological and physiological changes. In previous reports, fibroblasts have been found in the dermis layer and are related to skin aging and skin rejuvenation. Various growth factors and cytokines can induce the proliferation and migration of skin cells [19,20]. In this study, PPE was nontoxic to NHF and HFDPC cells but trended to decrease the cell viability of HaCaT cells at a high concentration. Cell viability higher than 100% may be associated with the cell-proliferation effect of PPE. PPE at high concentrations could stimulate cell growth in a time-dependent manner. Proteins and growth factors in the extract are important factors for the stimulation of cell proliferation [21]. Cell migration is expected to be associated with the re-epithelialization process of skin rejuvenation [22]. Recently, the TGF-β1was chosen as a positive control to promote cell migration of human skin fibroblasts [23]. In this study, PPE was able to induce the migration of human skin fibroblast cells in a time-dependent manner.

Oxidative stress and inflammation have been reported for aging progression if the redox system is not balanced [24]. ROS in the skin can stimulate the inflammation process, induce skin erythema, and affect the viability of skin cells. It also affects regulatory functions such as cell proliferation, cell senescence, or cytokine synthesis [2,25,26]. ROS can be generated in keratinocytes by specific processes (enzymatic oxidations and aerobic respiration, including several cytokines, etc.). Keratinocytes are essential sources of cytokines and lipid mediators, which are correlated to skin inflammation [3]. In our study, we used an H_2_DCFDA as a fluorescent probe, which has been widely used for the determination of intracellular ROS in living cells because this method is effective in cell loading with the cellular permeant fluorogenic probe, there is oxidation of the fluorogenic probe by cellular ROS, it is stable, there is retention of the fluorescent probes inside the cells, it is effective at detection, and easy to use [27]. PPE could reduce intracellular ROS production (at PPE concentrations of 10–2000 µg/mL). For the cutaneous inflammatory metabolism process, lipoxygenase enzymes control the metabolism of unsaturated fatty acids in three processes—oxidation of the substrates, transformation of hydroperoxy lipids, and synthesis of leukotrienes (LTs)—as the triggers of inflammatory reaction [4]. In this study, we evaluated anti-inflammatory activity via anti-lipoxygenase and measured the production of intracellular inflammatory cytokines. For lipoxygenase reaction, LA as a substrate is the most abundant PUFA in the epidermis, and it is rich in sphingolipids and ceramides, which are essential in maintaining the barrier and preventing trans-epidermal water loss [5]. Moreover, LOX enzymes are an essential source of ROS, and ROS is one of the oxidative stresses considered to be a component of inflammation. The LOXs are responsible for the metabolism of linoleic acid, in which peroxyl radical formation occurs during a LOX-catalyzed reaction. The inhibitor of LOX demonstrated anti-inflammatory and antioxidative against lipid peroxidation properties [28,29]. PPE could inhibit the lipoxygenase enzyme with a low IC_50_ value, signifying that PPE has potent antioxidant and anti-inflammation activity. For cytokines-induced inflammation, TNF-α is an essential proinflammatory cytokine in the human epidermis as it substantially impairs collagen synthesis and protection of the extracellular matrix [30]. PPE significantly reduced TNF-α production, which is concordant with recent research. Ahn et al. (2012) reported that protein hydrolysate had anti-inflammatory activity via inhibition of the production of TNF-α [31].

Hair aging and hair loss is commonly recognized as a sign of aging in mammals. Humans may exhibit premature hair loss. Human hair loss is an age-related condition whereby age causes defects in the hair growth cycle that are characterized by a rise in the ratio of the hair follicles in the telogen phase to those in the anagen phase caused by a reduced anagen phase and extended telogen phase [6]. Hair follicles are composed of epithelial components and dermal components. Dermal papilla cells (DPCs) are located in the hair bulb and affect the morphological characteristics and re-entry of the hair growth cycle. Therefore, DPCs were exploited as models to evaluate the effects of active substances in hair-growth-modulating studies [32]. The number of DPCs in the follicle is associated with the hair’s size and shape and the formation of a new hair shaft [33]. Furthermore, the expression of DPCs was associated with the aggregation behavior of the cells. The papilla cells can initiate hair development by the condensation of mesenchymal cells at the proximal end of the hair follicle [34]. As observed from the effect of PPE on cell viability and cell proliferation in DPCs, PPE was found to be nontoxic and could increase the proliferation of the cells in a dose-dependent manner. Furthermore, PPE had a high concentration (1000 µg/mL) of aggregation of DPCs in the aggregation test. Therefore, PPE may be a promising active compound for promoting hair growth.

To confirm the efficacy of this extract, an in vivo human study was performed. The PPE niosome serum demonstrated efficacy in producing the moisturizer effect by enhancing the water content in the stratum corneum. Aging conditions influence skin hydration. As seen in the result of % skin hydration, the formulation enhanced the skin hydration after treatment for 1 month [35,36]. For pigmentation and vascularity, the melanin index (pigmentation) and erythema index (vascularity) were interpreted as *%MI* and *%EI*. These values can indicate the changes in the skin based on light absorption in the principle of reflectance spectrophotometry [37]. The increase in *%MI* referred to high melanin content from the new hair follicles, which corresponds to the increase in %hair elongation, suggesting hair growth. *%MI* and % hair elongation were significantly increased after applying the serum for one month. Skin erythema arises when the skin comes into contact with irritants, for example, allergens and chemical substances. To assess the skin irritant indicator, *%EI* was measured [38]. After applying the formulation for one month, *%EI* significantly reduced from day 0. Hence, the serum containing 1000 µg/mL PPE is effective for skin and hair aging treatment and causes a low amount of irritation to the skin.

## 5. Conclusions

The bioactivities of PPE as an anti-aging of skin and hair were investigated. The extract could induce cell proliferation, cell migration, intracellular ROS reduction, lipoxygenase inhibition, and decrease intracellular TNF-α production in HaCaT cells. In addition, PPE increased cell proliferation and enhanced the cell aggregation of hair cells. Niosomes containing PPE in the form of the serum formulation demonstrated effectiveness in improving skin conditions in human volunteers. Therefore, the PPE-nanocarrier-loaded serum is effective for repairing skin and hair deterioration caused by aging.

## Figures and Tables

**Figure 1 pharmaceutics-14-01846-f001:**
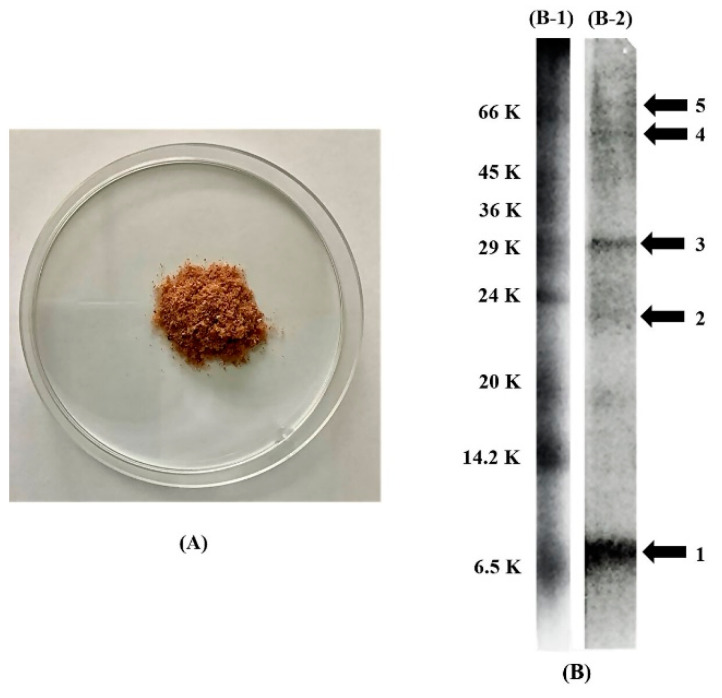
(**A**) Appearance of PPE extract and (**B**) the silver staining proteins on SDS-PAGE gel of protein markers (**B-1**) and unknown polypeptides of PPE (**B-2**).

**Figure 2 pharmaceutics-14-01846-f002:**
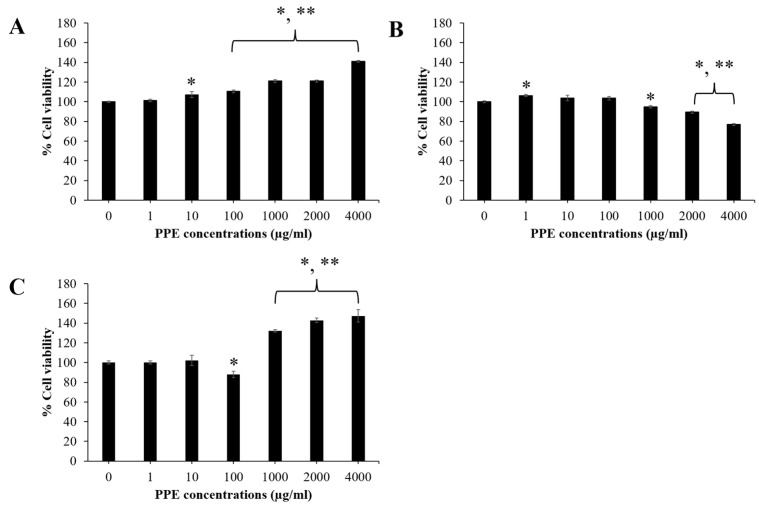
Effect of PPE on cell viability of (**A**) NHF; (**B**) HaCaT; (**C**) HFDPC. The cells were exposed to various concentrations (0–4000 µg/mL) of PPE for 24 h. Data are shown as mean ± SD (*n* = 3). *, ** represent a significant difference from DMEM group (a negative control) at *p*-values of <0.05 and 0.001, respectively.

**Figure 3 pharmaceutics-14-01846-f003:**
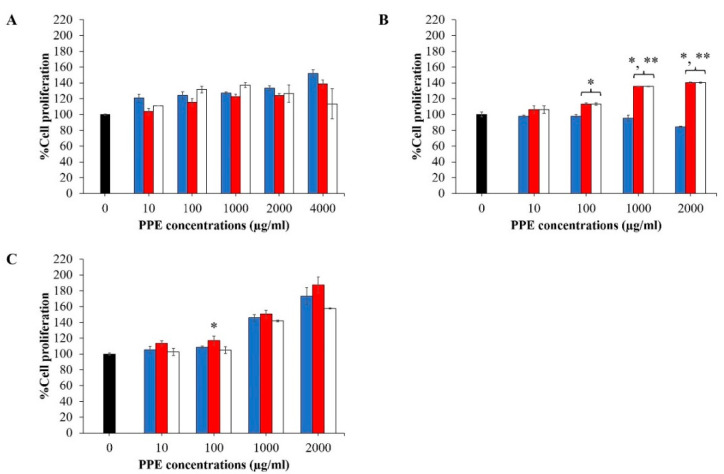
Effect of PPE on the proliferation of (**A**) NHF, (**B**) HaCaT, and (**C**) HFDPC cells. Each cell was treated with various concentrations (0–2000 µg/mL) of PPE for different time intervals: 24 h (■), 48 h (■), and 72 h (☐). Data are presented as mean ± SD (*n* = 3). * and ** represent the significant difference in % cell proliferation compared with the cell proliferation at 24 h at *p*-value < 0.05 and 0.001, respectively.

**Figure 4 pharmaceutics-14-01846-f004:**
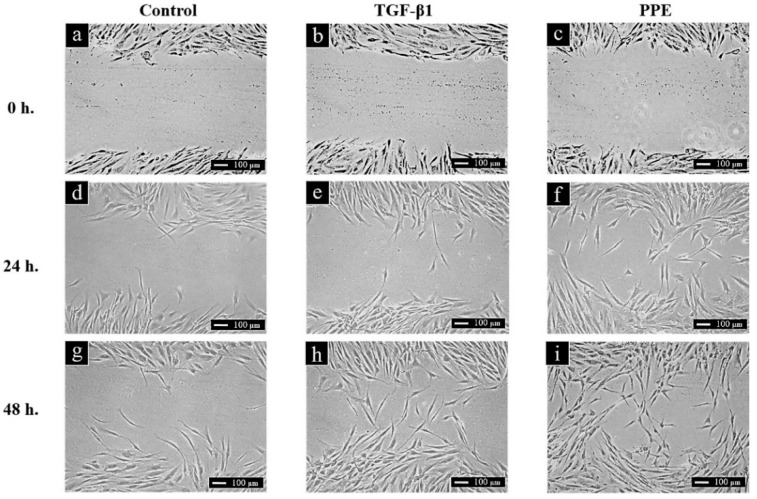
The width of the scratch area of control NHF cells (untreated cells) (**a**,**d**,**g**), TGF-β1 treated NHF cells (**b**,**e**,**h**), and 1000 µg/mL PPE treated NHF cells (**c**,**f**,**i**) at 0, 24, and 48 h after treatment.

**Figure 5 pharmaceutics-14-01846-f005:**
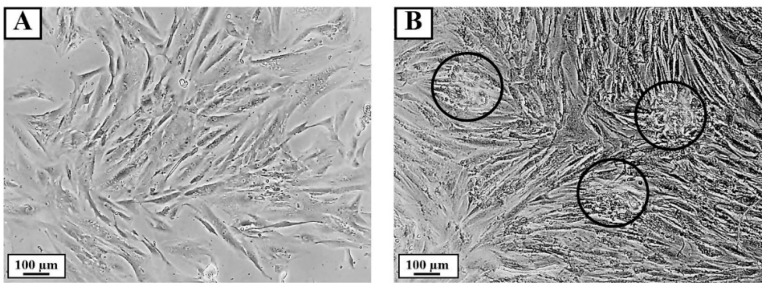
Effect of PPE on the aggregation of HFDPC cells after treatment for 72 h. HFDPC cells were treated with (**A**) DMEM (control group) and (**B**) 1000 µg/mL PPE. The images were photographed using an inverted microscope (10× objective lens, bright field).

**Figure 6 pharmaceutics-14-01846-f006:**
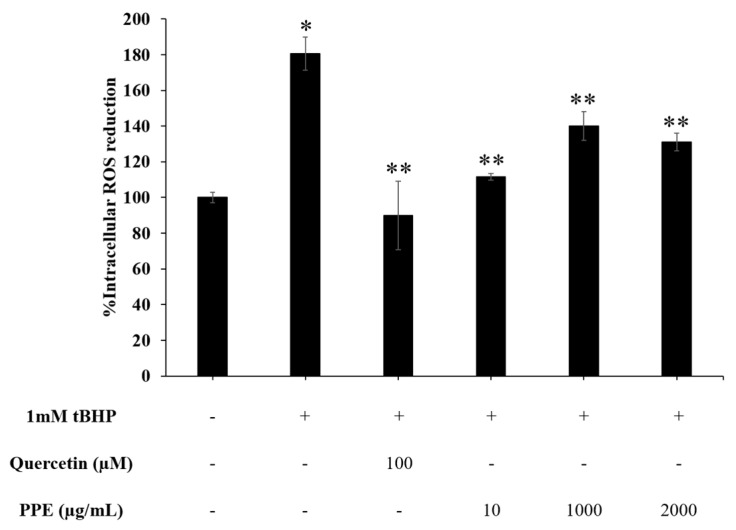
Effect of 100 µM quercetin and various concentrations of PPE (10–2000 µg/mL) on intracellular ROS reduction in HaCaT after induction of ROS production by 1 mM tBHP for 1 h. The DMEM-treated group was used as a negative control, and the t-BHP-treated group was used as a positive control. * and ** signify a significant difference (*p* < 0.05) when compared to the DMEM-treated group and the t-BHP-treated group, respectively.

**Figure 7 pharmaceutics-14-01846-f007:**
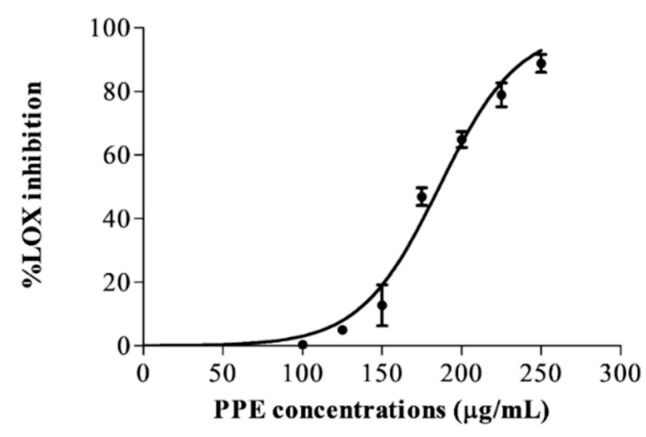
The % LOX inhibition of various concentrations of PPE (0–250 μg/mL).

**Figure 8 pharmaceutics-14-01846-f008:**
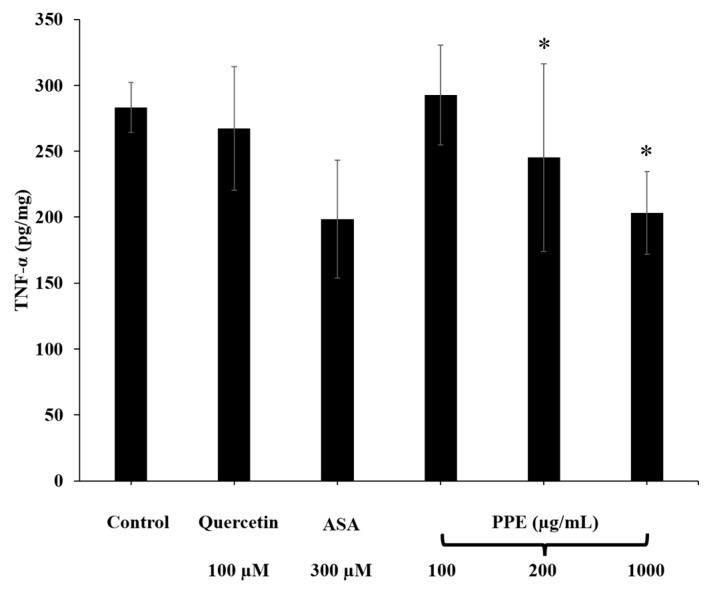
Inhibitory activity of 100 µM quercetin, 300 µM ASA, and various concentrations of PPE (100–1000 µg/mL) on TNF-α production in HaCaT cells. Data are expressed as mean ± SD (*n* = 3). * indicates a significant difference compared to control (*p* < 0.05).

**Figure 9 pharmaceutics-14-01846-f009:**
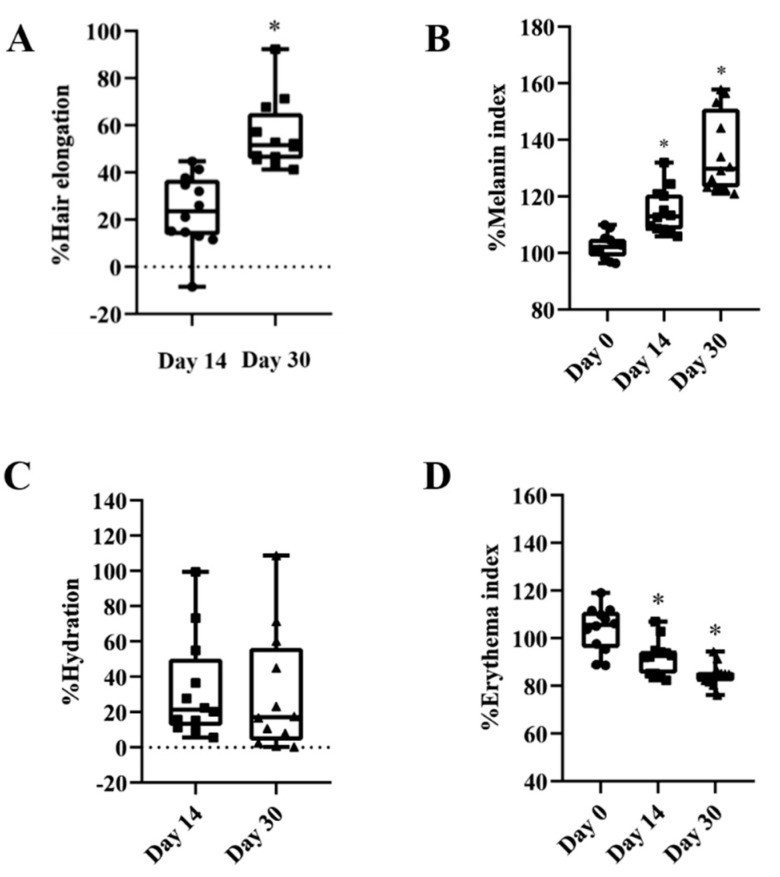
% Hair elongation (**A**), *%MI* (**B**), % hydration (**C**), and *%EI* (**D**) of the skin after application of the serum formulation in human volunteers for different periods. Data are reported as mean ± SD (*n* = 12). * signifies a significant difference from day 0 (*p* < 0.05).

**Table 1 pharmaceutics-14-01846-t001:** Characterization of niosomal solution and niosomal serum containing 1000 µg/mL PPE.

Formulations	Size (nm)	PDI	Zeta Potential (mV)
Niosomal solutions	107.60 ± 0.20	0.274 ± 0.02	−35.73 ± 1.50
PPE-loaded niosomal solutions	124.63 ± 2.02 **	0.352 ± 0.01 **	−32.0 ± 2.02
PPE-loaded niosomal serum	171.50 ± 1.08 *^,^**	0.328 ± 0.02 **	−28.5 ± 4.02 *

Data are displayed as mean ± SD (*n* = 3). The symbols * and ** signify a significant difference when compared to the niosomal solution with and without extract loading, respectively (*p* < 0.05).

## Data Availability

Not applicable.

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
