# Peer review of "In Vitro Biological Activity and In Vivo Human Study of Porcine-Placenta-Extract-Loaded Nanovesicle Formulations for Skin and Hair Rejuvenation"

_pharmaceutics, 2022, doi:10.3390/pharmaceutics14091846_

Round 1
Reviewer 1 Report
The manuscript described the mode of action of PPE on inflammatory/cell proliferation. Although they report findings on human, most of their manuscripts in performed in vitro, thus I find the title misleading. No factors measured in vitro has been assessed or evaluated in vivo limiting the value of this manuscript. Furthermore, the discussion fails to draw conclusion and is more akin to a introduction/method section. I also have several comments questions listed below:
Major Comments:
- Would like to see SDS page of PPE extract.
- Some growth factor porcine ELISA kit were not found. Detection limit of ELISA for growth factors?
- Define name of cell lines (or what type of cells they are) and why you used those. Also, why some test were performed with multiple cell lines, but not others (Fig 1: all cell types; then only NHF, then only HaCat, ...)?
- Figure 2: The graph is confusing; x-axis should be time and then graphs 1 curve for each concentration. What is the cell proliferation of controls at 48h vs 24h? Why is PPE toxic to HaCat at 24h but not 48h?
- Figure 4: what biological relevance/equivalence of HFDPC aggregation? what in vivo process does it mimic?
- Figure 5: why is PPE more effective at lower dose? What is the effect of PPE without tBHP?
- Figure 6: do you have a sham control (protein extract of non active proteins) to confirm that LOX inhibition is specific?
- Figure 7: Baseline level of TNF secretion seems high, and the positive control tended to be lower that control, which question the validity of you results.
Author Response
Comments:
- The manuscript described the mode of action of PPE on inflammatory/cell proliferation. Although they report findings on humans, most of their manuscripts in performed in vitro, thus I find the title misleading. No factors measured in vitro have been assessed or evaluated in vivo limiting the value of this manuscript. Furthermore, the discussion fails to draw a conclusion and is more akin to an introduction/method section.
- Thank you for your suggestion, we revised the title to “In vitro biological activity and In vivo human study of porcine placenta extract-loaded nanovesicle formulations for skin and hair rejuvenation”
- Would like to see the SDS page of the PPE extract.
- According to your comment, we have added the SDS page of the PPE extract in Figure 1.
- Some growth factor porcine ELISA kits were not found. The detection limit of ELISA for growth factors?
- In this study, we only quantified the amount of 4 growth factors (EGF, IGF-1, TGF-β1, and FGF) in PPE using the ELISA kit as these 4 growth factors are reported to be the main growth factors in the PPE.
- Define the name of cell lines (or what type of cells they are) and why you used those.
- In this study, three types of cells were used because we would like to study whether PPE can improve skin and hair rejuvenation. Normal human foreskin fibroblast cells (NHF) are representative of dermal fibroblast cells; human keratinocytes cells (HaCaT) are representative of keratinocyte skin cells, and human follicle dermal papilla cells (HFDPC) are representative of dermal papilla cells that are located at the bottom of the hair. These cells were defined in the material section.
- Also, why some tests were performed with multiple cell lines, but not others (Fig 1: all cell types; then only NHF, then only HaCaT)?
- Cytotoxic compounds can cause cell death. Therefore, cytotoxicity tests were investigated in multiple cells to ensure the low toxicity of PPE on 3 normal cells (skin and hair follicle cells).
- Skin regeneration and hair inductivity processes slow down with increased age. Skin regeneration is a natural cycle that occurs as the skin cells turnover and it is also associated with the replacement of damaged or dead cells with new tissue. A proliferation test of skin cells can be employed to determine the ability of a compound to boost skin regeneration. For hair inductivity, dermal papilla cells play an important role in regulating hair growth including hair formation, growth, and cycling. A proliferation test of hair follicle dermal papilla cells could confirm the ability of a compound to induce hair rejuvenation. Therefore, the ability of the PPE to induce skin and hair cell proliferation was investigated in skin and hair cells to describe skin and hair rejuvenation.
- Skin cell mobility generally involves two primary cell types in the skin (human keratinocytes (HaCaT) and human fibroblasts (NHF)). Biological aging can be expressed by the functional decline in cell migration of human dermal fibroblasts. Moreover, t-BHP was reported to induce ROS production and TNF-α stimulation on both keratinocytes and fibroblasts. However, these effects on the skin fibroblast cells (NHF) did not occur in our study. Therefore, only the results of these experiments on HaCaT cells were reported.
- For the aggregation assay, it is normally used to determine the ability of papilla cells to stimulate hair regeneration. Therefore, only human follicle dermal papilla cells (HFDPC) were used in this study.
- Figure 2: The graph is confusing; the x-axis should be time and then graphs 1 curve for each concentration. What is the cell proliferation of controls at 48h vs 24h? Why is PPE toxic to HaCat at 24 h but not 48 h?
- Thank you for your suggestion, we changed Figure 3 into a bar graph for better understanding.
- Control groups of 24 h and 48 h are cells treated with serum-free DMEM.
- PPE was toxic to HaCaTs at 24 h, but it was not toxic at 48 h because PPE has a proliferation effect on these cells.
- Figure 4: what biological relevance/equivalence of HFDPC aggregation? what in vivo process does it mimic?
- The formation of dermal papilla (DP) is related to the condensation of mesenchymal cells at the proximal end of the hair follicle, which determines hair shaft size and regulates matrix cell proliferation and differentiation. DP has the ability to regenerate new hair follicles. Several molecular markers have been used as an indicator of hair inductiveness, many of which are also associated with aggregation behavior.
Ref.: Sari AR, Rufaut NW, Jones LN, Sinclair RD. Characterization of ovine dermal papilla cell aggregation. Int J Trichology. 2016 Jul-Sep;8(3):121-9. doi: 10.4103/0974-7753.188966.
- In humans, HFDPCs are a cell population located in the bulge of the hair follicle with unique characteristics such as aggregative behavior and the ability to induce new hair follicle formation. In this study, hair follicle formation induced by the PPE could be observed by hair elongation and %MI.
- Figure 5: why is PPE more effective at a lower dose? What is the effect of PPE without tBHP?
- In a previous study, the concentrations (5-15 µg/ml) of PPE effectively decreased ROS generation in DCFDA/H2DCFDA- cellular ROS assay. However, the effect was not investigated at higher concentrations of PPE. Our results are in concordance with the study showing that a low concentration of PPE (10 µg/mL) could effectively reduce ROS production. However, increasing the PPE concentrations up to 1,000 µg/ml and 2,000 µg/ml did not potentiate the effect.
Ref.: Nensat C, Songjang W, Tohtong R, Suthiphongchai T, Phimsen S, Rattanasinganchan P, et al. Porcine placenta extract improves high-glucose-induced angiogenesis impairment. BMC Complementary Medicine and Therapies. 2021;21(1):66.
- The effect of PPE without t-BHP was not investigated because t-BHP was used to induce ROS production. Without t-BHP, cells will produce ROS production as normal and the amount of ROS in these cells will be calculated as 100%.
- Figure 6: do you have a sham control (protein extract of non-active proteins) to confirm that LOX inhibition is specific?
- We did not perform a sham-controlled study for the LOX inhibition effect.
- Figure 7: The baseline level of TNF secretion seems high, and the positive control tended to be lower than the control, which questions the validity of your results.
- TNF-α secreted from the non-treatment group is the baseline level of TNF secretion for the study. Quercetin and aspirin (ASA) (the positive control) have antiinflammation activity. When the cells were treated with these compounds, the level of TNF-α was decreased than that in the non-treatment group. Therefore, it is normal for the positive controls (Quercetin and ASA) to have a lower level of TNF than the non-treatment control.
Reviewer 2 Report
The authors use the MTT method to measure cell proliferation, which is a surrogate assay. the authors should be directly counting the cells, measuring cell proliferation with BrdU or EdU as a measure of cell prolifeation.
please rephrase the statement . ""In our study, PPE" on line 380
Author Response
- The authors use the MTT method to measure cell proliferation, which is a surrogate assay. the authors should be directly counting the cells, measuring cell proliferation with BrdU or EdU as a measure of cell proliferation.
- Per your comments, the MTT colorimetric assay is an established method of determining viable cell numbers in a proliferation study. MTT assay is also one of the popular techniques for quantification of measuring cell proliferation because it is reliable, rapid, and convenient.
Ref.: 1. Sylvester PW. Optimization of the Tetrazolium Dye (MTT) Colorimetric Assay for Cellular Growth and Viability. In: Satyanarayanajois SD, editor. Drug Design and Discovery: Methods and Protocols. Totowa, NJ: Humana Press; 2011. p. 157-68.
- please rephrase the statement. "In our study, PPE" on line 380.
- Thank you for your suggestion, the sentence has been rephrased as “In this study, PPE was able to induce migration of human skin fibroblast cells in a time-dependent manner.”
Reviewer 3 Report
The manuscript is interesting and well-designed. I have just a few comments:
Abstract
- Please present the main problem of the study.
Introduction
- Enter information that supports the use of the swine species for this purpose.
Materials and Methods
- The manuscript shows that porcine placentas were used. What is the origin of placentas? (please include information about the breeder, age of the animals, breed, and sanitary conditions.); The placenta was collected after delivery? The herd vaccinated and tested for infectious diseases? Has the project been authorized by the ethics committee in the use of animals? Present legal authorization or waiver statement.
Author Response
Reviewer #3
- Abstract: Please present the main problem of the study.
- Per your comments, the main problem of the study was included in the abstract
- The information “Porcine placenta extract (PPE) is containing many water-soluble macromolecular compounds, such as proteins and growth factors, which are limited transportation through the skin.” was added to the abstract.
- Introduction: Enter the information that supports the use of the swine species for this purpose.
- We added the information that supports the use of the swine species for this purpose in the text, lines 93-96 and Ref 16.
- Materials and Methods: The manuscript shows that porcine placentas were used. What is the origin of placentas? (please include information about the breeder, age of the animals, breed, and sanitary conditions.); The placenta was collected after delivery? The herd vaccinated and tested for infectious diseases? Has the project been authorized by the ethics committee for the use of animals? A present legal authorization or waiver statement.
- The porcine placenta was given by Charnchai Farm, Ratchaburi, Thailand. It was collected from large-white x landrace hybrid sows aged 2 years after a natural delivery. The pig was vaccinated following the recommended vaccination program and was grown in standard sanitary conditions. Currently, there is no need for approval from the ethics committee for using animal placenta that is obtained naturally after delivery. The placenta was obtained as part of routine animal production on a farm and the animal was not killed to get the placenta.
